# Engineering the Tumor Immune Microenvironment through Minimally Invasive Interventions

**DOI:** 10.3390/cancers15010196

**Published:** 2022-12-29

**Authors:** Koustav Pal, Rahul A. Sheth

**Affiliations:** Department of Interventional Radiology, The University of Texas MD Anderson Cancer Center, Houston, TX 77030, USA

**Keywords:** tumor microenvironment, interventional oncology, immunomodulation, locoregional therapy, CAR-T cell therapy, immunoengineering, interventional radiology

## Abstract

**Simple Summary:**

Interventional radiology is a highly evolving field that can modulate the various barriers imposed by the immunosuppressive tumor microenvironment. This review aims to showcase the various immune biophysical barriers that limit anti-cancer therapy by the tumor microenvironment and how interventional radiology possesses the facilities to overcome these barriers. These tools involve both physical and immune therapies that can be intratumorally injected to act locally but recruit a systemic response to produce a more potentiated anti-cancer therapeutic response.

**Abstract:**

The tumor microenvironment (TME) is a unique landscape that poses several physical, biochemical, and immune barriers to anti-cancer therapies. The rapidly evolving field of immuno-engineering provides new opportunities to dismantle the tumor immune microenvironment by efficient tumor destruction. Systemic delivery of such treatments can often have limited local effects, leading to unwanted offsite effects such as systemic toxicity and tumor resistance. Interventional radiologists use contemporary image-guided techniques to locally deliver these therapies to modulate the immunosuppressive TME, further accelerating tumor death and invoking a better anti-tumor response. These involve local therapies such as intratumoral drug delivery, nanorobots, nanoparticles, and implantable microdevices. Physical therapies such as photodynamic therapy, electroporation, hyperthermia, hypothermia, ultrasound therapy, histotripsy, and radiotherapy are also available for local tumor destruction. While the interventional radiologist can only locally manipulate the TME, there are systemic offsite recruitments of the immune response. This is known as the abscopal effect, which leads to more significant anti-tumoral downstream effects. Local delivery of modern immunoengineering methods such as locoregional CAR-T therapy combined with immune checkpoint inhibitors efficaciously modulates the immunosuppressive TME. This review highlights the various advances and technologies available now to change the TME and revolutionize oncology from a minimally invasive viewpoint.

## 1. Introduction

The tumor microenvironment (TME) is a dynamic landscape composed of multiple cellular and acellular components that impose barriers to anti-cancer therapies and promote tumor progression. The concept of the TME was first developed in 1863 when Virchow proposed that “lymphoreticular infiltrate” could point to the origin of cancer [1]. Our understanding of the TME has progressed substantially since that time. Currently, a common approach to conceptualize the TME is to discretize it into six unique niches: hypoxic, acidic, innervated, immune, metabolic, and mechanical niches [2]. Furthermore, it is now known that the TME comprises several key elements, including immune cells, stromal cells, and extracellular components. These niches and components engage in several complex interactions that aid tumor progression and metastasis. They also inhibit the therapeutic effects of conventional chemotherapies by excluding the localization of these drugs into the tumor and inducing tumor cell quiescence to diminish susceptibility to cytotoxic drugs. 

The TME also plays a crucial role in suppressing the anti-tumor efficacy of immunotherapies. Immune checkpoint inhibitors have revolutionized cancer care, but most patients do not exhibit an objective response to these therapies. Cancer cells can evade the immune system through numerous hurdles imposed by the TME. If these hurdles are able to be overcome, a tumor-specific adaptive immune response could be sparked within one site of disease, leading to a possibility that this spark could ignite a patient-wide immune response. In other words, by virtue of the capabilities of the adaptive immune system, it is known that local interventions can affect systemic tumor control. Thus, there is substantial interest in local interventions that can alter the tumor immune microenvironment to overcome immunosuppressive barriers and drive adaptive tumor immunity [3,4]. A great degree of creativity has been applied to engineering local interventions, utilizing technologies including ultrasound, radiation, photodynamic therapy, nanotechnology, electrotherapy, and thermal-based modalities [2,5]. This review summarizes the recent advances in engineering approaches to modulating the tumor immune microenvironment to stimulate tumor immunity.

## 2. The Tumor Microenvironment: An Overview

The tumor microenvironment consists of the tumor cells, stromal cells, immune cell infiltrates, blood vessels, extracellular fluid, and other extracellular elements. Several niches implicated in the TME include hypoxic, acidic, immune, mechanical, metabolic, and innervated niches. Discretizing the TME into individual niches is solely to facilitate conceptualization, as all niches are integrally involved in cross-talk and mutual interactions.

The hypoxic and acidic compartments are generated by intratumoral hypoxia, often due to the insufficiency of angiogenesis to provide for the increased metabolic requirement and cell turnover of cancer cells [6,7]. Hypoxia-inducible factor-1 (HIF-1) is one of the primary genes upregulated in hypoxia and leads to vessel proliferation via vascular endothelial growth factor (VEGF). Studies have shown that hypoxia allows signaling pathways enabling cancer cells’ invasiveness to spread disease [8,9]. hypoxia also stimulates p38/MAPK signaling, promoting cell migration of bordering cells for the epithelial to mesenchymal transition and, subsequently, metastasis [9,10]. Prolonged hypoxia in fibroblasts produces a stiffer extracellular matrix, a TME feature that can support the migration of breast cells and decrease the intratumoral deposition of systemically administered drugs [11]. Studies have shown that various TME hypoxia landscapes can be scored with a unique hypoxia signature, a collective of 15 hypoxia-associated genes [12]. Various cancer types are associated with differing scores, with increased hypoxia correlating with poorer prognosis [12,13,14]. Hypoxia has also been shown to induce cell cycle arrest and inhibit apoptosis while upregulating cellular chemoresistance. This affects drug delivery and efflux pumps and inhibits cytotoxicity required by chemotherapeutic drugs for their mechanisms of action [14,15,16]. 

Hypoxia also contributes to resistance to radiation therapy in tumor cells, often called the “oxygen effect”. Hypoxia increases the immunosuppressive environment by promoting M2 polarization in macrophages and attracting immunosuppressive T cells to the TME by changing the cytokine profile [17,18]. Drugs targeting HIF, vessel normalization, and supplemental oxygenation could reverse these changes and allow for better tumor regression. Several clinical trials involving immune checkpoint inhibitors with anti-VEGF agents are ongoing to prove these hypotheses (NCT02366143, NCT02684006, and NCT02853331) [13,19,20,21]. The vascular normalization hypothesis suggests that by correcting abnormal angiogenesis through anti-angiogenic therapies, there can potentially be a reversal of abnormal TME, leading to better drug penetration and tumor cell death [7,19,22,23,24]. The TACTICS trial supported this in vivo by pretreating HCC patients with sorafenib, following up with TACE as required. Sorafenib is a tyrosine kinase inhibitor that exhibits anti-angiogenesis and anti-proliferative properties by acting at RAF, VEGFR, and PDGFR [25,26]. The group that had pretreatment with sorafenib and TACE had a more prolonged progression-free survival than those treated only with TACE (25.2 months vs. 13.5 months alone) [27].

The acidic niche develops through hypoxic and metabolic alterations to the tumor microenvironment, which eventually promotes tumor survival, growth, migration, invasion, and glycolysis. In fact, dysregulated pH is a known hallmark of tumors [28]. Due to hypoxia and the irregular vasculature of the tumor, cancer will have highly hypoxic and acidic areas due to poor perfusion. As a result, increased expression of various ion transporters and enzymes such as carbonic anhydrase become upregulated to achieve tumoral homeostasis [29]. This leads to a phenotypic change of tumoral survival that aids in progression, immune surveillance escape, and resistance to therapy. Acidosis is often a selection pressure for tumor cells, where those that develop preferential adaptations survive and proliferate, avoiding regulated cell death [29,30]. Studies reversing tumor acidosis with buffering to restore pH in a B-cell lymphoma have shown that there was a promotion of IFN-γ expression in NK cells, switching to an immunostimulatory microenvironment and allowing for more immune cell invasion within the TME [28,30]. 

Metabolic changes in the tumor microenvironment are also partly due to several genetic changes involving lactate metabolism, reactive oxygen species (ROS), and lipids. The Warburg effect states that tumor cells prefer anaerobic glycolysis to aerobic glycolysis, even in oxygen-rich states. This eventually generates lactate, which lowers the TME pH. Lactate is a valuable byproduct for TME to reprogram immune macrophages from M1 to M2 phenotypes while promoting the survival of the immunosuppressive Tregs within the TME. It also promotes angiogenesis. Glucose and glutamine are consumed in this process, and studies have shown that glutamine utilization by tumor cells remodels the TME, inducing hydroxylation of collagen to make a stiffer, firmer extracellular matrix for the TME [24,31,32]. HIF-1 upregulates lactate dehydrogenase to produce higher lactate levels in tumoral cells, which generates onco-metabolites and further perpetuates the acidotic TME. Knockouts of LDH inhibit cell growth and increase tumor cells’ radiosensitivity. AT-101 is a novel non-selective inhibitor of LDH, decreasing prostate cancer tumor load [33]. More selective LDH inhibitors are being investigated to target TME acidosis and lactate metabolism.

## 3. Biophysical Barriers of the TME

The TME is a tailor-made landscape for tumorigenesis. The tumor secretes various factors which enable the TME to be shaped and crafted into the ideal environment for cancer progression [34,35]. Due to increased extracellular matrix density (ECM), the transport of drugs to the tumor interstitium is impaired. High vessel permeability, as well as the tumor mass itself, leads to vessel compression [7,22]. Solid stress on the surrounding area due to the tumor contributes to the extravasation of fluid outside the tumor [36,37] leading to high interstitial fluid pressure [22]. Increased interstitial fluid pressure (IFP), dense ECM and intratumoral lymphatic vessel collapse lead to decreased pressure gradients within the tumor and surrounding vasculature. This decreases the effective blood and oxygen supply to the tumor, exacerbating the hypoxic and acidic environment. This is due to switching to a lactate-based metabolism [22,38]. Studies have also demonstrated that this induces cancer-associated fibroblasts to undergo a desmoplastic reaction, to produce more collagen, and increase the amount of hyaluronan and sulfated glycosaminoglycans [31]. All these self-perpetuate the tumor stiffness and increase the component of biophysical stress throughout the TME [39]. Furthermore, the increased IFP and stiffness of the tumor influence drug delivery [40]. Increased pressure causes fluid to leak out of vessels supplying the tumor, while impairing lymphatic drainage of TME. Tumors are poorly perfused as a result, while stiffer ECM decreases diffusion of drugs to reach the tumor cells. The net IFP gradient hence increases drug washout from the tumor [41] While chemotherapeutic agents can kill tumor cells in a Petri dish, mechanical barriers like this cause failure to deliver chemotherapeutic agents to reach the tumor, contribute to systemic toxicity, and perpetuate tumor progression [8,19,22]. Hence, if the attenuation of TME reduces mechanical stress and therapeutic drug washout, there will be more effective tumor cell killing and reduced chemotherapeutic toxicity. Several proposed drugs reduce fibrosis generated by the CAF, which includes anti-VEGF agents to repair leaky vessels. Even non-oncologic drugs such as losartan can act as anti-fibrotic agents to remodel the TME. Losartan demonstrated an anti-collagen effect to remodel the TME, reducing stress and allowing for better drug penetration [42,43].

## 4. Local Therapeutic Interventions to Modulate the Tumor Microenvironment

Delivering systemic therapy to treat a localized tumor often is met with a wide range of systemic toxicities which do not effectively kill the tumor cells. As such, local therapies that can eradicate tumors in patients with solitary or oligometastatic disease are the standard of care for many cancer types. However, a burgeoning role for local anti-cancer therapies is in the local immunomodulation of a small fraction of lesions in patients with metastatic disease. Given that local immunomodulation can potentially stimulate systemic tumor immunity, there is a growing rationale for pursuing such interventions for their immunologic rather than their cytotoxic capabilities [44]. These interventions can be broadly categorized into therapeutic local delivery systems involving nanoparticles and devices, local drug delivery, and molecular targeting. In addition, physical methods involve photodynamic therapy, radiotherapy, ultrasound, thermal-based therapies, and electrical therapies. 

### 4.1. Local Delivery of Immunotherapeutics 

Compared to conventional intravenous administration of treatment drugs throughout the body, local intratumoral drug delivery allows for a higher local concentration. Given the recent renaissance in immunotherapies, intratumoral immunotherapy trials have resulted in a resurgence. Recent studies have demonstrated that variables from needle design and drug formulation alongside immunotherapy can influence intratumoral drug delivery’s efficacy. Multisided hole needles led to approximately threefold improvements in intratumoral drug deposition in a mouse model, while using longer-acting STING-loaded MDP hydrogels prevented excess drug extravasation in the same tumor model [40,45]. This showed decreased adverse effects as well. Hydrogel-embedded nanoparticles in a glioblastoma mouse model showed a similar increase in tumor-specific distribution and long-term retention of the nanoparticles. Hydrogels are particularly efficacious in tumors where elevated IFP leads to drug washout [46].

### 4.2. Nanomedicine

The field of nanomedicine is extremely vast and out of the scope of this review [19,47,48,49,50,51,52]. We aim to briefly discuss some of the available nanoparticles available from an oncological point of view. Nanoparticles (NP) are particles with one dimension less than 100 nm with unique properties that are not found in bulk samples of the same material [53]. Most are based on the same basic structure (Figure 1): the surface layer, shell layer, and core. The basic design is then modified to suit a particular purpose. Nanoparticles are often grouped into polymer and liposome-based drug delivery systems. Nanoparticles have deep tumor penetration and longer retention, and modifications such as PEG help to decrease immune system clearance [53,54]. To optimize delivery to TME, nanoparticles are engineered with the ability to be stable in the blood, escape immune support, and reach the TME with high-pressure penetration [37]. Nanoparticles available for current oncological use include nab-paclitaxel (Abraxane), which is used for breast, pancreatic, and non-small cell lung cancer treatment. Liposomal forms of chemotherapy include doxorubicin hydrochloride, which is commercially available as Caelyx and Myocet; these are used for the treatment of breast cancer, multiple myeloma, ovarian cancer and Kaposi’s sarcoma [55]. Benefits of nanoparticle-based Abraxane decreased the previously associated hypersensitivity reactions with its previous generation of drug known as Cremophor (which was paclitaxel combined with polyoxyethlated castor oil) [56]. Hence, nanoparticles have unique properties due to their size and administration, leading to a varied number of positive benefits [57].

Nanoparticles can be selectively targeted with either passive or active targeting systems. Passive systems use the leaky, abnormal vessels of the TME to have enhanced permeation and retention of the nanoparticles. Examples of NP that use passive targeting include Genexol PM, which is paclitaxel, and a sterile lyophilized micellar formulation [58]. It allowed for a three times maximum tolerated dose in nude mice. Other examples include Abraxane, and DaunoXome (liposomal daunorubicin) [37]. 

Active targeting involves exploiting tumor-specific ligands binding to nanoparticles to cause a downstream effect. For example, HER2-targeted PEGylated liposomal doxorubicin has reduced cardiotoxicity [59]. Nanoparticles to target hypoxia by silencing HIF-1a with siRNA have also been developed [60,61]. 

Nanorobots are a particular type of nanoparticle which can convert power sources into kinetic energy [62]. These can be biohybrid systems, chemically powered, or physically powered [62]. Nanorobots for medicine are mainly developed with the aim to either deliver a certain payload, or to manipulate micro-objects [63,64].

Nanorobots can deliver drugs with higher precision and speed than simplistic nanoparticles, as they do not entirely rely on passive diffusion. Nanoparticles are not as site-specific as nanorobots, as the reticuloendothelial system and organs uptake some nanoparticles [65,66] in contrast, nanorobots can be directed and targeted towards a particular site [62]. New nanorobots coated with magnets can be directed toward a particular site with magnetic fields [53,67].

Nanorobots could potentially be the future for a minimally invasive mode of therapy. Nanorobots can be programmed to act only at a specific site to perform various functions that target TME. The TME is an acidic, hypercoagulable state due to overexpression of tissue factor [19,47,48]. One such nanorobot has been constructed with DNA origami technology to cause selective thrombosis by being equipped with a truncated tissue factor which causes tumor-specific thrombosis. Newer models have used nanoparticle-based thrombin, which activates only in the presence of tumor nucleolin to induce thrombosis. This model was efficacious in melanoma and ovarian cancer mouse models [49,50]. Other nanorobots can be equipped with chemotherapeutic agents such as doxorubicin and utilize the acidic microenvironment and hydrogen peroxide of the TME to drop their payload. Catalyzing the hydrogen peroxide into water and oxygen also helps to alleviate the hypoxia of TME as well [51]. A unique feature of nanorobots is that there are no reported systemic toxicities nor any immunologic side effects compared to CAR-T cells [52].

### 4.3. Sustained Release Biomaterials

Drug depots are implantable devices that elute a particular formulation of a drug locally or by drug delivery nanocarriers. Like nanoparticles, they can deliver the drug of interest within the tumor either by passive or active targeting [68,69,70]. These are efficacious because they allow for better local drug concentration with minimal systemic side effects, more so than nanoparticles, and allow for controlled temporal sustained release. The type of drug depot, either a monolithic matrix or a reservoir implant design, is chosen based on the drug kinetics [71]. Zero-order drugs are implanted with a reservoir implant, while first-order kinetics are paired with a matrix implant. Combinations of chemotherapeutic, immunologic, and gene therapy can also be released simultaneously, triggered either by external stimuli such as Infrared or ultrasound or internal stimuli such as pH and body temperature [72,73]. Depots can alter the TME by also overcoming hypoxia. A proposed oxygen delivery depot showed better tumor cytotoxicity when combined with doxorubicin in a mouse model. Alginate pellets implanted in hypoxic regions of the mouse model themselves did not have any anti-tumor activity but effectively killed tumor cells with doxorubicin [74]. Biodegradable drug depots are currently being developed to prevent postoperative tumor recurrence. One such device has been loaded with gemcitabine with PEG in a pancreatic cancer mouse model and showed minimal systemic drug toxicity [75]. Studies are ongoing to refill local drug depots noninvasively [76].

### 4.4. Implantable Microdevices

Implantable Microdevices (IMDs) are miniature devices that can be implanted into a patient’s tumor percutaneously. They can release up to 20 drugs stored in micro-reservoirs via diffusion into spatially separate regions of a tumor. The device is then removed after three days for analysis, which can be histopathology, metabolomics, or multiplexed immunofluorescence [73,77]. Tatorva and colleagues have developed an implantable microdevice that could administer several combinations of chemotherapeutic and biologics in different spatially separated regions of the breast cancer mouse model [78]. When combined with multidimensional analysis and flow cytometry, the efficacy of tumor killing along with the immune response could be studied. This allows for the potential to give individualized treatment for particular tumors: implant the device, find synergistic combinations and provide the most effective treatment. Implantable microdevices are safe for patients with various tumors [73] and due to micro-dosing, there are no systemic toxicities. The microdevice was able to test multiple permutation combinations in the mouse model and demonstrated that Panobinostat, venetoclax, and anti-CD40 therapy showed a complete tumor remission [78]. 

## 5. Physical Therapies to Alter the TME 

Summary in Figure 2.

### 5.1. Photodynamic Therapy

Photodynamic therapy (PDT) involves a two-step process by which a photosensitizer, which can be oral, topical, or intravenously administered. After accumulation in the malignant tissue, around 24–72 h in the targeted tissue, light is given to the photosensitizer, producing ROS or singlet oxygen molecules to cause apoptosis, necrosis, or autophagy [79]. This generates an immune response, where released damage-associated molecular patterns (DAMPs) activate neutrophils and CD8+ T cells. This causes direct tumor toxicity and activates the complement system to boost PDT efficacy further. TNF-a, IL-6, and IL-1B are secreted to propagate the immune system effects [80,81]. PDT combined with immunomodulatory agents induces a sustained immune response, resulting in greater T-cell activation, switching TME from an immunosuppressive environment to an immunostimulatory environment, and showing effective rejection of tumor rechallenge [80]. The use of PDT as an adjuvant shows promise, and PDT combined with radiotherapy in breast cancer mouse models have shown to improve trabecular structure in bone metastasis. PDT usually only has side effects related to the skin, with no or minimal systemic side effects [82]. PDT is also cheaper than radiotherapy as well [82]. PDT requires dose-dependent modification as per the tumor site and is difficult to administer to deep tissues due to the failure of penetration of light to reach the photosensitizer [83]. Newer advances involve photo-immunotherapy, where conjugation of the photosensitizer with specific antibodies targeting cancer-associated antigens has been developed, allowing for selective immune-stimulatory responses against the target tumor tissue [79].

### 5.2. Electrical Therapies

#### 5.2.1. Electrochemotherapy

Electrochemotherapy (ECT) involves the injection of chemotherapeutic drugs either intravenously or intratumoral with local application of electrical impulses that enhance drug uptake [84]. ECT also enhances chemotherapy and nanomedicine’s local permeability and retention (EPR) effect [85]. Electroporation can lead to either reversible or irreversible structural membrane effects that can lead to an enhanced EPR effect [86]. ECT increases the uptake of poorly permeant chemotherapeutic drugs such as Bleomycin. Bleomycin and ten 500 V/cm pulses for the liver cancer mouse model were efficacious [87]. ECT has been proven for skin tumors and breast cancers, and ongoing clinical trials for gynecological, GI, and head and neck cancers are underway [85]. ECT used for cutaneous breast metastasis in a cohort of Italian patients showed a 64% complete response and was more effective when combined with immunotherapy than with chemotherapy [88].

#### 5.2.2. Irreversible Electroporation

Irreversible electroporation (IRE) only affects the cell membrane of the tumor cells, allowing for other components of TME to be unchanged. IRE is preferred for tumors close to blood vessels, such as liver and pancreatic cancer. IRE performed in eight patients of unresectable hilar cholangiocarcinoma had increased progression-free survival of 18 months [89]. IRE also showed similar efficacy in prostate cancer compared to standard radical prostatectomy in terms of 5-year recurrence and better-preserved urogenital function as it does not affect local architecture in stage-4 patients [82,87,88,90].

#### 5.2.3. Tumor Treating Fields

Tumor treating Fields (TTF) is a series of low- to intermediate-frequency alternating electric fields applied to the tumor using electrical applicators attached to a portable battery pack. This arrests the cell cycle only for dividing cells, eventually leading to cell cycle arrest and death [91]. TTF is currently approved for glioblastoma and mesothelioma. TTF interferes with tubulin alignment during mitotic spindle formation, which inhibits metaphase and telophase of the cell cycle [91]. Other anti-tumor mechanisms include increased drug uptake, anti-cell migration permeability, immunogenicity, and autophagy [90,92]. DNA repair is also impaired with increased DNA double-strand breakage. TTF showed a modest increase in progression-free survival in newly diagnosed GBM patients by 2.7 months compared to those treated with temozolomide with no other side effects [92]. Further trials are ongoing for recurrent ovarian cancer, unresectable gastric adenocarcinoma, as well as with adjuvant immunotherapies and checkpoint inhibitors to increase the abscopal effect [93,94]. (NCT04281576, NCT03477110, NCT03705351, NCT04221503, NCT03995667, NCT02973789).

### 5.3. Thermal-Based Therapies

Hyperthermia modulates the TME by causing head-based coagulative necrosis of the local target tissue. This involves microwave ablation, radiofrequency ablation (RFA), and high-intensity focused ultrasound (HIFU). When tumor cells are destroyed, DAMPs, PAMPs, and released nucleic acids enhance the immunostimulatory effect, increasing the adjuvanticity of tumor cell immunogenicity [95]. Furthermore, there was increased tumor perfusion and reoxygenation of the TME, switching the M2 to M1 phenotype of macrophages as well [96]. 

#### 5.3.1. Radiofrequency Ablation 

Radiofrequency ablation (RFA) is a minimally invasive form of thermal ablation. An electrode tip is placed into the tumor, where frictional heat and heat conduction produce three zones of hyperthermic injury: a central necrotic zone, peripheral sublethal hyperthermia, and the unaffected surrounding zone. The TME experiences an inflammatory infiltrate in the peripheral area, while the central necrotic zone produces various DAMPs that further upregulate the immune response [95,96,97]. Jiang et al. have demonstrated a difference in direct RFA (due to thermal contact from the electrode) compared to indirect RFA(due to thermal transfer away from the electrode) and that tumors are more prone to resistance in direct heating compared to the effects of indirect RFA [97]. This bears caution to the thermal impact of RFA, as some studies have reported a 25–39% increase in distant new tumors in patients being treated for HCC [98,99] combination of RFA with drugs, such as c-Met inhibitors (c-Met is involved in several signaling pathways such as PI3k/AKT, Ras/MAPK, JAK/STAT, and VEGF) could block distant metastasis of liver cancer [99,100,101]. 

#### 5.3.2. Microwave Ablation (MWA)

MWA is dielectric tissue heating caused by oscillating tissue water molecules in alternating electromagnetic fields. MWA is different from RF heating in that MWA heats the area around the applicator, whereas RFA must be conducted through areas of current for thermal transfer [102]. Recent studies have described abscopal effects of MWA where after ablation of HCC in 23 patients, there was a moderate increase in circulating immune cells after day seven following ablation, with higher proportions of effector memory T cells. Six patients also demonstrated tumor antigen-specific IL-5 [103]. This shows the potential of MWA to elicit abscopal recruitment and potentially lead to tumor immunity as well. A recent case study demonstrated an increased immune response with MWA in an acquired immune-resistant squamous non-small cell lung cancer patient who was treated with PD-1 and VEGFR-TK inhibitors. CT-guided images before and after the MWA demonstrated a gradual decrease in lymphadenopathy and other lobe tumors after the primary tumor was ablated [104]. MWA for breast cancer in 35 patients resulted in 32 patients having complete ablation of the tumor, with an increase in immune CD4+ T cell and IFN-γ compared to surgically resected tumors (n = 13) [105]. 

#### 5.3.3. Cryoablation (CRA)

Cryoablation is tissue destruction by freezing [106]. The cryoablation-induced injury occurs through a cycle known as the freeze–thaw cycle. This cycle leads to cell death by physical ice-related damage to cells, protein denaturation, necrosis, activation of apoptosis, and immune recruitment [107,108,109,110]. Cell destruction leads to the activation of anti-tumor immunity due to the release of tumor-related antigens such as SNAP23 and STXBP2, leading to an abscopal effect. In a mouse melanoma model, cryoablation increased effector cells and decreased Treg cells and M2 macrophages [111]. A recent study compared RFA to Cryotherapy ablation in a colon cancer mouse model, where the authors suggested that RFA/heat-based ablations have a more significant anti-tumoral effect than CRA [112].

### 5.4. Ultrasound-Based Therapies

Ultrasound therapies are promising for modulating TME and tumor treatment. Two ultrasound modes can modulate TME: (a) low-intensity pulsed ultrasound (LIPUS) and (b) High-intensity focused ultrasound (HIFU) [113,114]. HIFU refers to intensities greater than 5 W/cm^2^, which produces both mechanical and thermal effects based on the duration and intensity given at a particular point. Adjustable HIFU beams for a specific tumor stiffness achieve the required heating or automatic ablation level. While thermal ablation causes coagulative necrosis, mechanical ablation does not induce as much surrounding thermal damage [113]. HIFU has demonstrated the ability to alter various gene expressions within the TME by increasing tumor cell apoptosis. This was seen with higher expressions of HSP-70, BCL-2, BAX, BAD, and Bak [115]. M-HIFU was recently combined with immune checkpoint blockade in triple-negative breast cancer mice models, and this showed more potent tumor growth suppression, immunostimulatory to M1 phenotype, and increased T-cell CD8+ population compared to conventional HIFU [116]. MR-HIFU is a variant of HIFU combined with magnetic resonance and allows for more specific temperature and site specificity regulation [117]. MR-HIFU was combined with anti-PD1 checkpoint inhibitors in mice with multi-focal breast cancer [117]. An increased global inflammatory response was observed, with upregulation of *Nod1*, *Nlrp3*, *Aim2*, and other innate immune receptors. A meta-analysis of whole gland HIFU for prostate cancer compared to radical prostatectomy showed the 5-year treatment-free survival to be higher amongst the HIFU population [118]. Furthermore, fewer side effects such as urinary incontinence and impotence were observed than in robot-assisted laparoscopic prostatectomy [119]. A trial of HIFU in breast cancer patients demonstrated a significant decrease in immunosuppressive cytokine levels of TGF-B1, IL-6, and IL-10 after HIFU treatment [120].

Histotripsy is a non-thermal focused variant of ultrasound therapy that uses microsecond (cavitation) or millisecond (boiling) pulses to cause cavitation bubbles that lead to tumor ablation [121]. The rapid expansion and collapse of these bubbles lead to the mechanical destruction of the target tissue. Histotripsy ablation in various mouse models of pancreatic and hepatic models has shown significant increases in survival, with the same benefits of immune recruitment as compared to M-HIFU. Furthermore, there are no thermal ablation side effects compared to conventional HIFU [122,123]. Histotripsy and other ultrasound methods are noninvasive and tissue-selective as it preferentially damages tumors compared to more elastic tissues (such as vessels, ducts, bowel, and nerves), have a minimal transition zone, avoid bleeding risk, and stimulate an immune response [114,120,124]. Histotripsy cannot be used on gas-containing organs, has a chance of thrombosis, and requires high ultrasound pressures, which may not be deliverable in harder-to-access, deeper tissues [120,121,125,126]. A phase 1 trial of hepatic histotripsy in inoperable liver cancer was conducted in Spain (NCT03741088). Out of 11 tumors in 8 patients, 10 had local tumor regression after two months, with 2 patients having a continuous decrease in relevant biomarker activity in HCC and CRC metastasis [127]. The THERESA trial was the first-in-human trial of hepatic histotripsy in unresectable multi-focal HCC or unresectable liver mets from various cancers. Eight patients all achieved planned tissue destruction with no device-related adverse effects. Furthermore, two patients had an increased abscopal effect with non-targeted tumors also decreasing in size after the procedure [128].

Sonoporation is a technique enabling ultrasound waves to cause stabilized gas microbubbles to oscillate, potentially opening up vascular barriers and allowing for the extravasation of a drug in a specific location [129]. There are two types, high-frequency, which has thermal effects, and low frequency, which has stable cavitation and non-thermal effects [130]. A gemcitabine-based microbubble study for inoperable pancreatic cancer showed promising results, where there was a significant increase in median survival (8.9 months to 17.6 months *p* = 0.011) with minimal extra side effects compared to controls [129].

### 5.5. Radiotherapy

Radiotherapy is administered to up to 50% of patients for cancer patients. There is extensive data on the immunologic ramifications of radiotherapy, and a comprehensive review of this field is beyond the scope of this manuscript. However, we will discuss the effect of radiotherapy on the TME and some of the latest advances in radiotherapy [131]. Radiotherapy effectively causes direct and indirect damage to the tumor cells and TME. Direct DNA damage, protein, collagen fragmentation, and cross-linkages occur. Indirect actions such as ROS production, telomere shortening, and biomolecule oxidation eventually lead to cell death [3,131,132]. In the post-radiated TME of glioblastoma, radioresistance forms lead to further pro-angiogenic signaling, increased stemness, and immune invasiveness, leading to increased chances of tumor recurrence [131]. Advances in radiotherapy to prevent resistance include a combination of immune checkpoint inhibitors and developments such as hypofractionated RT and FLASH-RT. These involve giving higher doses of radiation for a lower duration and a more specific site. 

The organ-specific TME also influences the treatment response. A study by Yu and colleagues demonstrated that cancers that have liver mets induce apoptosis of CD8+ T cells primed to the tumor, leading to the generation of an immune desert phenotype. Furthermore, this is resistant to immune checkpoint inhibitors. This was demonstrated in multiple mouse models, comparing models with primary tumors and those with induced liver mets. Mice without liver mets responded to immunotherapy, while those with metastasis did not. When radiotherapy was combined with the current modality, the TME became sensitive to immunotherapy again [133,134].

While conventional radiotherapy relies on the “4Rs”—radiation, repopulation, repair, and redistribution, newer methods give higher doses to directly cause cell necrosis along with increased DAMPs and PAMPs upregulating the abscopal effect [135,136]. Hypofractionated RT has demonstrated that there are increased anti-tumoral immune responses and increased immunostimulatory effect due to the increased radiation dose stimulating effect [137]. Several trials are ongoing to test the superiority of hypofractionated radiotherapy to conventional radiotherapy. However, as of now, studies have proved their “non-inferiority,” which is questionable [138]. 

Localized delivery of radiation for liver tumors has also been investigated. Transarterial radioembolization (TARE) involves the injection of radioactive yttrium-90 microspheres, which allow for higher radiation doses directly to the tumor. A recent study by Deiployi and colleagues demonstrated that TARE in breast cancer liver metastases demonstrated a heterogeneous pattern of changes to immune markers before and after TARE in 20 patients, including elevation in Il-10 and increased CD4+ tumor infiltration (15 vs. 31%, *p* < 0.001), with eight patients achieving complete response [139]. However, further studies are required to assess the utility of TARE vs. stereotactic RT and TACE [140,141].

## 6. Disrupting the Tumor Immunity Cycle through Interventional Immunoengineering

For most cancers, the TME is in an immunoinhibitory state due to various immunosuppressive factors being secreted, such as TGF-b, Il-10, and acidotic and hypoxic factors. So-called “cold” tumors are those whose TME exhibits minimal tumor-infiltrating lymphocytes and a subdued immune response, while “hot” tumors are those with localized, potentially inactivated TILs. In addition to being insensitive to immune checkpoint inhibitor therapy, cold tumors are associated with poorer prognosis [142]. Interventional procedures to modulate the TME can improve immunogenic recruitment and switch tumors from a “cold” to a “hot” environment. However, a tailored, biologically driven approach is necessary to overcome the specific barriers in any given TME. This can include therapies that affect (1) T-cell priming and activation, (2) T-cell expansion and T-cell infiltration.

### 6.1. T-Cell Priming and Activation

Various therapies can be administered to increase the number of activated T-cells, including oncolytic viruses, hyperthermic methods, chemotherapy, photodynamic therapy, and radiotherapy [142,143]. Oncolytic viruses show promise as they have a tropism for selective tumor infection and subsequent death while also releasing PAMPs and DAMPs, leading to further T-cell priming and activation [144,145]. However, IV administration of the viruses is complicated due to washout and backflow [146,147]. Using intra-arterial or direct tumoral delivery leads to enough pressure to overcome the increased IFP of the TME. Delta-24-RGD is a well-developed oncovirus that has been used in glioblastoma treatment. When combined with endovascular selective intraarterial therapy, there is an increased concentration reaching the tumor, decreased systemic toxicity, the potential for repeat delivery, and further penetration past the blood–brain barrier [144,148]. Intratumoral injection of OVs in 18 patients of nasopharyngeal carcinoma achieved improved median progression-free survival (29.6 vs. 8.4 weeks) [149]. 

### 6.2. T-Cell Expansion

The increased population of T-cells can be increased either with tumor vaccines or adoptive-cell therapies. Tumor vaccines are grouped into cell, peptide, viral or nucleic acid-based vaccines. Oncolytic viruses have already been discussed, but peptide and nucleic acid-based vaccines also show promise. Nucleic acid vaccines can code for multiple tumor antigens, which offers a robust immune response. Newman and colleagues have discovered that intratumoral injection of unadjuvanted seasonal influenza vaccines in melanoma mouse model increased the CD8+ T cell population and decreased Tregs in TME, effectively converting a “cold” tumor into a “hot” one [150]. Yellow fever vaccines in intratumoral mouse models with Immune-checkpoint inhibitors also demonstrated increased T-cell expansion of CD8+ T cells [151]. Adoptive cell therapies include CAR-T therapy and locoregional or intratumoral administration to allow for effective treatment. For example, intratumoral mRNA c-MET CAR-T cells injected in metastatic breast cancer patients were well tolerated and resulted in higher levels of immune infiltrates and tumoral necrosis. Two out of six patients also exhibited peripheral CAR-T cells without any signs of severe side effects [152]. The authors hypothesize that combining this mode of treatment with checkpoint inhibitors or STING agonists may further increase the immune response.

Adusumilli et al. have demonstrated that intrapleural injection of mesothelin-targeted CAR-T therapy for malignant pleural disease proved effective when combined with immune checkpoint inhibitors [153]. A phase 1 trial for malignant pleural mesothelioma in 18 patients with cyclophosphamide pre-conditioning and anti-PD1 treatment had a median OS (95% CI) of 23.9 months compared to 17.7 months (n = 23) of just cyclophosphamide treatment. This was compared to the median OS of cisplatin and pemetrexed first-line treatment OS of 13 to 16 months [154]. Cyclophosphamide pre-conditioning helps to dampen the immunosuppressive TME and recruit APCS for a more robust anti-tumor immunity [155]. Regional administration of CAR-T cells showed peripheral CAR-T presence within just four days for 87% of the patients and persistence in peripheral blood for up to 100 days in 39% of patients. This study is just a phase 1 trial but shows promise as it showed minimal systemic side effects, was well tolerated, and demonstrated persistence of the regional CAR-T cells when combined with pembrolizumab. The authors also hypothesize that the exogenous CAR-T cells benefitted not only with a PD-1 antagonist, but the endogenous T cells were also recruited to attack mesothelin negative tumor cells due to the presence of newer IgG responses. A phase II study with a fixed dose of mesothelioma CAR-T cells and pembrolizumab is ongoing [153].

## 7. Conclusions

Systemic delivery of immunotherapies has revolutionized cancer care. However, to broaden their impact across the cancer spectrum, adjuvant interventions are likely needed to overcome immunosuppressive TME. New bioengineering techniques such as implantable microchips, hydrogels, and nanoparticles are well-poised to address this unmet need. However, all treatments can potentially become a double-edged sword. For example, excess immunotherapy may enhance immune stimulation to kill tumors, reducing tumor recurrence rates, but at the cost of systemic toxicities or tumor growth if the immune titration between stimulatory and suppression is not met. A new chapter in tumor treatment by altering the TME is underway, and further research and relevant clinical trials involving local therapy and immunotherapy are necessary for minimally invasive interventions compared to the current standards of care.

## Figures and Tables

**Figure 1 cancers-15-00196-f001:**
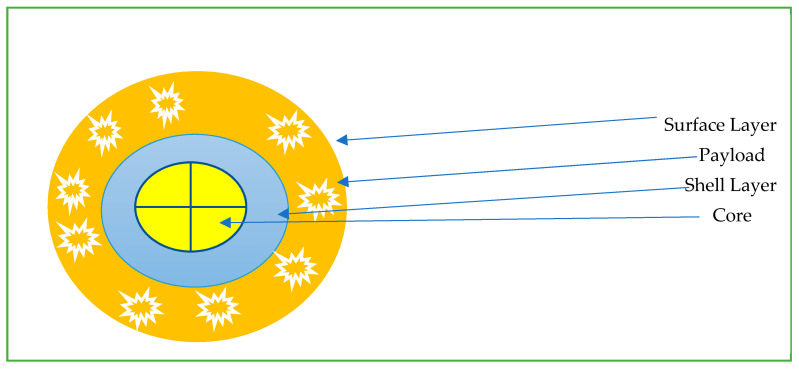
Common schema for nanoparticles for intratumoral delivery.

**Figure 2 cancers-15-00196-f002:**
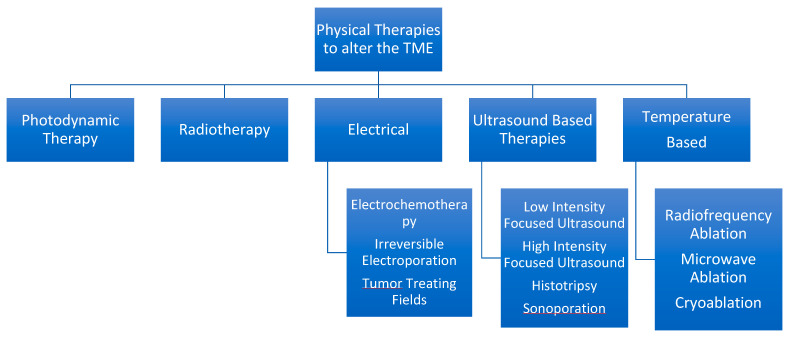
Hierarchal Chart to Describe various physical therapies available from a minimally invasive standpoint.

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
