# Peer review of "Engineering the Tumor Immune Microenvironment through Minimally Invasive Interventions"

_cancers, 2022, doi:10.3390/cancers15010196_

Round 1
Reviewer 1 Report
In their manuscript entitled “Engineering the Tumor Immune Microenvironment through 2 minimally invasive interventions”, Pal et al. sought to review the various biophysical barriers of the tumor microenvironment that limit anticancer therapy and how treatment strategies used in interventional radiology, such as local delivery systems and physical methods, may overcome these barriers by modulating the tumor immune microenvironment and stimulating anti-tumor immunity.
Overall, this review is well written and structured. It covers a clear medical need in oncology. However, there are some concerns/issues in the manuscript the authors should address before the article may be considered for publication in Cancers:
1. The addition of high-quality illustrations and/or figures would greatly enhance the quality of the article and help convey the main points in a visually appealing way.
2. The section on nanoparticles is too simplistic in its current form and does not the preclinical and clinical landscape of this therapeutic strategy in different cancers. For examples, nanoparticles are broadly classified into natural (e.g., extracellular vesicles) and synthetic (e.g., liposomes, PGLA, gold, etc. formulations), all of which have different inherent properties that are being explored for drug delivery. See van der Koog et al. 2021, Elsharkasy et al. 2020, and others.
Author Response
see attached document

Reviewer 2 Report
cancers-2058130 Review
Engineering the Tumor Immune Microenvironment through minimally invasive interventions
This Review paper is too brief in review content and boring in its discussion.
I think this manuscript difficult to accept this manuscript.
Author Response
see attached document

Reviewer 3 Report
This review article from Pal and Sheth describes the barriers of the tumor microenvironment and how they suppress productive antitumor immune responses. The review also discusses strategies to disrupt or overcome these barriers through physical ablative approaches or localized delivery of therapeutics. Overall, this is an important contribution to the literature although the impact may be greater if the authors focused on interventional/localized approaches, which are underreported in the literature, rather than systemic/targeted approaches such as nanomedicine for which there are endless publications with limited clinical efficacy.
Minor comments:
Line 26: add hypothermia or cryoablation to the list of physical therapies
Line 27: the meaning of “While we only modulate…” is not clear
Line 89: how is anoxia different from hypoxia?
Line 91: references needed to support that hypoxia promotes M2 polarization and immunosuppressive T cell recruitment
Line 100: it should be noted that sorafenib has other antitumor mechanisms beside anti-angiogenesis
Line 114: IFN-gamma?
Line 138: Please explain how mechanical stressors exacerbate hypoxia?
Line 144-147: It is not clear from the description how the leaky vasculature and fluid extravasation causes failure of chemotherapeutics to reach the tumor
Line 182: what distinguishes ‘nanorobots’ from ‘nanoparticles’ ?
Line 196: not “all” nanoparticles have the same structure
Line 216: clarify how nanoparticles are not as specific as nanorobots
Line 282: “cell membrane of the tumor” implies that a tumor has one membrane
Line 276: IRE, ECT and TTF should be subsections within “Electric field-based therapies”
Line 300: “adjuvant treatment, and mesothelioma” doesn’t make sense
Line 311: information about MWA should be added as is it a significant ablative technology
Line 311: information about cryoablation should be added as it is a significant ablative technology
Line 316: “there increased” does not make sense
Line 325: define direct vs indirect RFA
Line 335: “frequency” should be “focused”
Line 340: “genetic effects” is misleading
Line 385: The section on Radiotherapy is interesting in that a lot of clinical research has been done on its immunomodulatory effects, however, it seems out of scope. More detail on the IR approaches would bring more impact
Line 461: reference needed for the mRNA c-MET CAR-T cell study
Author Response
see attached document
